# Research on the Impact of Enterprise Digital Transformation on Internal Control

**Chenxi Wang [1], Deli Wang [2,*], Xincai Deng [1] and Shun Wang [3]**

1 School of Economics and Management, Guizhou Normal University, Guiyang 550001, China; 201512001@gznu.edu.cn (C.W.)
2 School of Accounting, Guangdong University of Foreign Studies, Guangzhou 510006, China
3 School of Business, Anhui University, Hefei 230601, China
* Correspondence: 202110039@oamail.gdufs.edu.cn

**Abstract:** Digital transformation is a crucial strategy for enterprises to achieve improvements in quality, efficiency, and dynamism. It represents the key direction for enterprise innovation and reform, and it is also of great significance for promoting high-quality economic development. While the previous research has shown that digital transformation can affect business efficiency, performance, and corporate governance, there is a lack of literature clearly linking digital transformation to internal control. To determine whether enterprise digital transformation impacts the constructions and implementation of internal control, we analyze data from Shanghai and Shenzhen A-share listed companies between 2012 and 2019. Our study found that the extent of digital transformation in enterprises has a positive impact on the establishment and effectiveness of internal control. Furthermore, we examine the role of market competition in the relationship between the digital transformation and internal control. We found that the impact of digital transformation on internal control is limited in enterprises with relatively mild market competition, while in more fiercely competitive markets, the positive impact of digital transformation on internal control is more pronounced.

**Keywords:** digital transformation; establishment of internal control; effectiveness of internal control; market competition

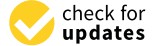



## 1. Introduction

Digital transformation refers to leveraging cutting-edge digital technologies to encourage enterprises to innovate their organizational structures and business models, thereby creating value, improving productivity, and enhancing social well-being [1]. Compared to traditional informatization, enterprise digital transformation involves the integration and application of digital technologies, such as cloud computing, big data, artificial intelligence, and blockchain in managing businesses. The thorough integration of these technologies into the real economy has made the digital economy a new growth point for economic development. Digital transformation is a process that involves using a range of emerging digital technologies to enhance enterprise value creation and adapt to changes in the external environment [2]. Digital technology empowerment is the core of digital transformation [3]. With the continuous development of digital technologies, such as artificial intelligence, blockchain, cloud computing, and big data, digital transformation has become a crucial direction for innovation and transformation for enterprises. Digital transformation's focus on digital empowerment offers enterprises new possibilities for production, performance, and technological progress. Additionally, its digital governance effects create new opportunities to address corporate governance issues [4,5]. Digital transformation not only provides vitality for sustainable business development, but also compels organizations to innovate in their organizational models and operational methods.

Through digital transformation, enterprises can innovate and enhance daily operations and management processes, achieving digital and intelligent development at the en-

terprise and industry levels. This is a critical approach for enterprises to achieve quality, efficiency, and power changes, which are significant for quality development. As a highly discussed topic in academia and the industry, the expectations for the digital transformation of enterprises is high. Many scholars anticipate that its coverage and penetration rate will rapidly increase [6]. On a positive note, the utilization of digital technologies can offer numerous benefits, such as providing more valuable information, reducing the costs of information search, offering multi-dimensional and visualized annotations, and enhancing the transparency of financial accounting information [7]. However, certain obstacles hinder the digital transformation of enterprises. The adoption of digital technologies may also create strategic risks, particularly regarding the inability to directly reflect its benefits of short-term strategic goals or financial indicators, which can impact managerial decision-making outcomes.

Internal control is a process, effected by an entity's board of directors, management, and other personnel, designed to provide reasonable assurance regarding the achievement of control objectives [8]. Previous research has shown that digital transformation can impact a company's operating efficiency, performance, information disclosure, and corporate governance [9–11]. Given that the purpose of internal control is to provide reasonable guarantees for the realization of a company's legal compliance, asset security, financial reporting, business, and strategic goals, it is clear that digital transformation will inevitably have an impact on the internal control of enterprises, to a certain extent. Therefore, it is important for companies to assess the impact of digital transformation on their internal control systems and implement the appropriate measures to ensure that their control objectives are effectively met. Nevertheless, there is a lack of empirical literature that explicitly establishes a direct link between digital transformation and internal control.

To determine whether enterprise digital transformation impacts the constructions and implementation of internal control, we analyze data from Shanghai and Shenzhen A-share listed companies between 2012 and 2019. Furthermore, we examine the role of market competition in the relationship between digital transformation and internal control.

We found that the digital transformation of enterprises has a significant impact on the establishment and effectiveness of internal control. Furthermore, the digital transformation of enterprises has mainly improved the establishment of internal control in areas, such as risk assessment, control activities, and information and communication. Additionally, the digital transformation of enterprises has improved the overall achievement and effectiveness of the enterprise, particularly in areas such as asset security, financial reporting, operating, and strategic objectives. It is also important to note that the positive impact of enterprise digital transformation on internal control may be limited in relatively moderate market competitions. However, when market competition becomes more intense, the benefits of digital transformation on internal control become more pronounced.

We make several important contributions in our study. First, unlike previous normative studies, we employ empirical methods to investigate the effects of digital transformation on the establishment and implementation of internal control, thus expanding the existing literature on internal control. Second, we explore the impact of specific digital transformation sub-items, such as artificial intelligence, cloud computing, and digital technology applications on internal control. This reveals the specific elements that have a significant relationship with internal control in the digital transformation of enterprises. Third, we examine the impact of digital transformation on the five elements of internal control and internal control objectives. This reveals how digital transformation can improve the establishment and effectiveness of internal control. Finally, we conduct an in-depth investigation into how the competitive market environment affects the relationship between digital transformation and internal control, and examine the heterogeneity of the impact of digital transformation on internal control in different market competition environments. Our findings hold significant implications for enterprises looking to enhance the establishment and implementation of internal control by leveraging the governing influence of market competition. Overall, we provide important insights into the relationship between

digital transformation and internal control, which can be used as a foundation for future research in this area.

## 2. Theoretical Analysis and Research Hypothesis

Although the study of internal control has been a popular area of research in academia, the regulatory requirements for internal control in different countries were generally lenient before the implementation of the SOX Act [12]. The lack of sufficient information disclosure during this period led to early research on internal control mainly focusing on the institutional analysis of the internal control system or evaluating its effectiveness through measures such as earnings revision [13], accounting errors [14], and restatement of financial statements [15]. The implementation of laws and regulations, such as SOX, which mandate the disclosure of internal control defects, has greatly advanced the research on internal control in various countries [16]. The existing literature has extensively discussed the factors that influence the effectiveness of internal control, covering three main aspects: the characteristics of the enterprise itself, internal governance factors, and external factors. The factors influencing internal control encompass almost all aspects of the internal and external operations of the enterprise [17–19].

Internal control is a process, effected by an entity's board of directors, management, and other personnel, designed to provide reasonable assurance regarding the achievement of control objectives [8]. Internal control comprises five interrelated and mutually supporting elements, namely, control environment, risk assessment, control activities, information and communication, and monitoring activities. These elements form a cohesive and logically linked system. Each element also serves the five objectives of internal control, which include ensuring legal compliance, safeguarding assets, ensuring accurate financial reporting, achieving business objectives, and supporting strategic objectives. Thus, the five elements of internal control work together to achieve these objectives.

Prior research has indicated that the digital transformation of enterprises will have a significant impact on various aspects, including operational efficiency, performance, information disclosure, and corporate governance, collectively impacting all elements of internal control [9–11].

First, digital transformation represents a process that affects the control environment of a company in various ways. It requires organizations to integrate and leverage advanced digital processing technologies, such as big data, IoT, and artificial intelligence, to enhance their organizational structure and innovate their business model [20]. As a result, the digitalization of a company leads to a shift in corporate strategy and triggers a series of internal environmental changes, including modifications to the organizational structure, human resource policies, and even corporate culture.

Second, digital transformation can enhance the efficiency and effectiveness of risk assessment [21], which is a prerequisite for implementing effective control activities. In order to conduct effective risk assessments, enterprises require timely access to adequate and appropriate information, as well as the ability to identify risk factors effectively. Compared to traditional information technology, new-generation digital technologies, such as big data, blockchain, and IoT, can acquire a wider range of information and data, and enable the real-time storage and recall of large amounts of data. Additionally, digital technologies, such as cloud computing and artificial intelligence, can accurately identify the risk factors related to business activities and establish effective predictive models, enabling more precise assessments of the probability of each risk factor occurring. Furthermore, these digital technologies can help decision makers select more effective and appropriate risk response strategies by correctly classifying the risk levels.

Third, digital transformation enables enterprises to overcome cost–benefit constraints and reduce the risk of fraud in human operations, thereby enhancing the effectiveness of control activities [22]. Additionally, in a digitalized environment, the control constraints on employees are uniform, thereby increasing the effectiveness of control activities.

Fourth, digital transformation has revolutionized the way enterprises access and use data, leading to improved management efficiency and increased value of information. According to Fenwick and Vermeulen [11], digital technology enables enterprises to access more information and leverage it to analyze and predict the choices and behaviors of board members, thereby reducing agency costs and monitoring expenses for public enterprises. Additionally, digital transformation enhances communication efficiency, enabling the real-time sharing of information among stakeholders, and facilitating collaborations and coordinations within the organization. In addition, digital transformation enables enterprises to optimize their organizational structure and improve their efficiency of information transmission, as well as their innovation potential [10]. With the advancement of cloud computing, companies can, at present, centralize, coordinate, and share financial tasks, leading to the optimization and streamlining of financial processes. Additionally, digital transformation enables companies to efficiently process and report financial information and provide sufficient information to external stakeholders. In his study, Ivaninskiy [23] discovered that the use of blockchain technology can enhance shareholder voting participation and improve the effectiveness of internal controls by providing smart contracts.

Fifth, with regard to monitoring activities, the implementation of digital technologies in companies can significantly reduce the costs associated with supervision. The adoption of big data and cloud computing technologies for auditing, as well as the use of distributed ledger technology, such as blockchain, have led to the development of more intelligent and comprehensive audit systems. Moreover, the blockchain's tamper-evident and transparent nature can improve the existing regulatory mechanisms by providing more accurate and reliable records of financial transactions and other related data. As a result, these emerging technologies are changing the way we approach regulatory and audit practices in the corporate world [24].

From a business management perspective, digital transformation has significantly enhanced business efficiency by promoting innovation, reducing transaction costs, and improving resource allocation efficiency. Enterprises generate vast amounts of data during their production and operational processes, and effective digital transformation is essential to efficiently process both internal and external information with the help of technologies, such as cloud computing, big data, artificial intelligence, and the Internet of Things. By doing so, enterprises can improve the utilization and value of their data and make informed decisions that drive business success. In particular, the integration of digital technologies can help businesses reduce operational costs and increase productivity by streamlining processes, improving supply chain management, and automating routine tasks. As a result, digital transformation is critical to remaining competitive in today's rapidly evolving business landscape [25]. Enterprises can leverage big data and artificial intelligence technologies to make more informed business decisions and achieve their goals. Digital transformation is critical to building the necessary infrastructure to harness the power of these technologies, enabling enterprises to strengthen innovation momentum at the operational level and increase investments in the research and development. This, in turn, expands the digital transformation process and improves operational efficiency, allowing enterprises to achieve better operating performances even under resource constraints. Furthermore, digital transformation provides a platform for enterprises to gain a competitive edge in the market by enhancing their ability to rapidly respond to changing market demands and shifting customer preferences. As such, digital transformation is a crucial strategy for businesses aiming to thrive in today's fast-paced and rapidly evolving business environment [9].

According to Acemoglu [26], integrating digital technology into traditional business models and optimizing production resources through technological innovation is essential for companies seeking to maintain a competitive edge. In particular, leveraging data as a key factor of production can significantly impact a company's business model and drive innovation-driven transformations, thereby enhancing its competitive advantage.

A comprehensive review of the existing research on the digital transformation of enterprises reveals that such transformations have had a significant impact on driving technological advancement, as well as on achieving corporate goals in terms of operational efficiency, performance levels, information disclosure, and corporate governance. In light of this review, we propose two hypotheses related to the impact of digital transformation on internal control:

**Hypothesis 1:** *The extent of the digital transformation of enterprises will have a significant positive impact on the establishment of internal controls when other conditions remain constant.*

**Hypothesis 2:** *The extent of the digital transformation of enterprises will have a significant positive impact on the effectiveness of internal controls when other conditions remain constant.*

## 3. Research Design

### 3.1. Data Sources and Sample Selection

We utilized data from Shanghai and Shenzhen A-share listed companies between 2012 and 2019 as the initial research sample. To ensure the data's reliability and accuracy, we excluded incomplete data or those from the financial and real estate industries, as well as specially treated samples. All continuous variables were winsorized at the 1% and 99% levels to reduce the effects of outliers.

The data related to internal controls were obtained from the DIB internal control and risk management database, which was the first professional and authoritative internal control information database in China. This database provided an objective and accurate reflection of the internal control levels of Chinese listed companies. We obtained financial data for other listed companies from the CSMAR database and annual report data from the official websites of Shenzhen and Shanghai Stock Exchanges.

### 3.2. Variables' Design

#### 3.2.1. Dependent Variables

The mandatory disclosure of information on internal control deficiencies made the identification and disclosure of material weaknesses a crucial aspect in evaluating the effectiveness of an enterprise's internal control. However, the lack of detailed criteria for identifying internal control deficiencies in China's "Guidelines for the Evaluation of Enterprise Internal Control" provides enterprises significant discretion in this process. This discretion, coupled with opportunistic tendencies, such as seeking advantages and avoiding disadvantages, cost–benefit trade-offs, and other factors, can make it challenging for enterprises to accurately determine the severity of internal control deficiencies. This situation may result in some material weakness or significant deficiencies not being identified or disclosed, potentially undermining the effectiveness of internal control.

We analyzed data from the "Blue Book on the Implementation of Enterprise Internal Control Regulation System of Listed Companies", disclosed by the Ministry of Finance and the China Securities Regulatory Commission. Specifically, we examined the number of internal control deficiencies disclosed by listed companies that published internal control evaluation reports in China between 2012 and 2019 (Table 1), as well as their internal control evaluation conclusions (Table 2).

Our analysis focused on two aspects: the disclosure of material weaknesses and the effectiveness of internal control evaluations. We observed that, on average, 2.11% of listed companies in China disclosed material weaknesses, with the lowest value of 0.35% in 2012 and the highest value of 3.82% in 2019. Additionally, the average proportion of listed companies with invalid internal control evaluations was 0.53%, with the lowest value of 0.13% in 2012 and the highest of 1.16%. Comparing these results with those of listed companies in the United States, we found that the proportion of effective internal control evaluations in China was significantly higher. However, these findings also support the previous re-

search indicating that the disclosure of material weakness in Chinese enterprises may be underestimated or concealed.

**Table 1.** Disclosure of internal control deficiencies by listed companies.

| Year | Number of Companies Disclosing Material Weaknesses | Number of Companies Disclosing Significant Deficiencies | Number of Companies Disclosing Control Deficiencies | Number of Companies Disclosing No Control Deficiencies | Total |
|---|---|---|---|---|---|
| 2012 | 8 | N/A | N/A | 2236 | 2244 |
| 2013 | 31 | 37 | 377 | 1884 | 2312 |
| 2014 | 39 | 53 | 455 | 2047 | 2571 |
| 2015 | 36 | 57 | 817 | 1809 | 2678 |
| 2016 | 42 | 40 | 895 | 1989 | 2930 |
| 2017 | 69 | 51 | 890 | 2282 | 3245 |
| 2018 | 123 | 74 | 1084 | 2256 | 3456 |
| 2019 | 139 | 38 | 360 | 3132 | 3642 |

**Table 2.** Conclusions on the effectiveness of internal control of listed companies.

| Year | Number of Companies with Effective Internal Control (%) | Number of Companies with Invalid Internal Control on Non-Financial Report-Related Deficiencies (%) | Number of Companies with Invalid Internal Control on Financial Report-Related Deficiencies (%) | Number of Companies with Invalid Overall Internal Control (%) | Others (%) | Total (%) |
|---|---|---|---|---|---|---|
| 2012 | 2241 (99.87%) | N/A | N/A | 3 (0.13%) | N/A | 2244 (100%) |
| 2013 | 2287 (98.92%) | 9 (0.39%) | 8 (0.35%) | 6 (0.26%) | 2 (0.09%) | 2312 (100%) |
| 2014 | 2538 (98.72%) | 9 (0.35%) | 16 (0.62%) | 6 (0.23%) | 2 (0.08%) | 2571 (100%) |
| 2015 | 2649 (98.92%) | 7 (0.26%) | 16 (0.60%) | 6 (0.22%) | 0 (0%) | 2678 (100%) |
| 2016 | 2898 (98.91%) | 10 (0.34%) | 18 (0.61%) | 4 (0.14%) | 0 (0%) | 2930 (100%) |
| 2017 | 3177 (97.90%) | 13 (0.40%) | 33 (1.02%) | 21 (0.65%) | 1 (0.03%) | 3245 (100%) |
| 2018 | 3342 (96.70%) | 26 (0.75%) | 48 (1.39%) | 40 (1.16%) | 0 (0%) | 3456 (100%) |
| 2019 | 3513 (96.46%) | 21 (0.58%) | 72 (1.98%) | 36 (0.99%) | 0 (0%) | 3642 (100%) |

Note: Other situations are as follows: 2 listed companies did not issue internal control evaluation conclusions in 2013; 2 listed companies had material weaknesses in 2014, but they did not distinguish between financial and non-financial reporting deficiencies; 1 listed company was unable to determine the effectiveness of its internal control system in 2017.

The DIB internal control and risk management database developed two internal control indices in response to concerns about the disclosure of internal control deficiencies, taking into account the institutional background of China. The first index was goal-oriented and constructed based on the five major objectives of internal control, enabling the evaluation of the degree of realization of internal control objectives. This index reflects the internal control level and risk management capabilities of the listed companies.

The second index was based on the five elements of internal control. The DIB internal control and risk management database constructed this index using information disclosed by listed companies about their internal control systems. From the perspective of the control environment, risk assessment, control activities, information and communication, and supervision activities, this index measures the establishment and improvement of the com-

pany's internal control system, reflecting the degree of perfection of the internal control establishment of listed companies.

The indexes constructed by the DIB internal control and risk management database comprehensively measure the internal control level and risk management capability of listed companies in China, objectively reflect the operating status of the internal control of listed companies, and provide a quantitative basis for the scientific research on internal controls. Some studies use the information disclosure-oriented internal control index as an indicator to measure the quality of internal control information disclosure. Through an in-depth analysis of the construction process of information disclosure-oriented internal control index, it can be found that it did not pay attention to the disclosure behavior of internal control information, but the information content in the disclosed information. The DIB internal control and risk management database clearly pointed out that the purpose of constructing this index was to measure the level and quality of internal control and risk management and control the capabilities of enterprises. Referring to Lu [27] and Cheng [28], we used the natural logarithm of the information disclosure-oriented internal control index as the proxy variable of internal control establishment (Establishment), aiming to measure the establishment and soundness of the internal control of enterprises from the perspectives of control environment, risk assessment, control activities, information and communication, and supervision activities. At the same time, we used the natural logarithm of the goal-oriented internal control index as the proxy variable of internal control effectiveness (effect), aiming to measure the ultimate efficiency and effect of the implementation of enterprise internal control.

### 3.2.2. Independent Variable

At present, there are three types of methods used to measure the extent of digital transformation in enterprises. The first method, as used by He [29], involved assessing whether an enterprise underwent digital transformations in the current year and used this as a binary variable. However, this approach has limitations in effectively measuring the extent of digital transformations and may introduce bias to the measurement. The second method, employed by Qi [30] and Zhang [31], measured the extent of digital transformation by using the proportion of digital economy-related intangible assets disclosed in the notes to financial reports of listed companies relative to total intangible assets. The third method, as used by Wu [32] and Yuan [33], involved conducting a textual analysis of the annual reports of listed companies and measuring the degree of digital transformations based on the presence of keywords related to digitization. The concluding and instructive sentences in a company's annual report not only reflect its strategy and goals, but also reveal the management's business philosophy. In our research, we measured the frequency of these relevant words in the reports as an indicator of the extent of digital transformation of the companies.

The CSMAR database at present provides direct access to the data on the extent of digital transformation among enterprises. These data are generated by extracting the text content of annual reports of A-share listed companies on the Shanghai and Shenzhen Stock Exchanges and using it as a basis for screening feature words. By searching, matching, and tallying the frequency of these characteristic words, the frequency of key technical terms was classified and collected to form a total word frequency and then construct an index of the extent of the digital transformation for each enterprise. The feature word lexicon was mainly derived from the previous research on the digital transformation literature and important policy documents. Negative words, such as "no", "not", and "non-", were excluded before the keywords.

### 3.2.3. Control Variables

We referred to the previous research related to the determinants of internal control [17,19,34] and included the appropriate control variables in Table 3.

**Table 3.** Definitions of the main variables.

| Variables | Definition |
|---|---|
| Dependent Variables | |
| Establishment | ln(information disclosure-oriented internal control index) |
| Effect | ln(goal-oriented internal control index) |
| Independent Variable | |
| DT | The log of total frequency of technical terms in key directions counted using text analysis methods |
| Control Variables | |
| Size | The log of a firm's total assets at the end of the fiscal year |
| Age | The log of the number of years the firm has listed |
| Sub | Number of subsidiaries in a fiscal year |
| Lev | The ratio of total liability to total assets |
| Sales | The log of operating income in a fiscal year |
| Cash | The ratio of cash and cash equivalents to total assets |
| Rec | The ratio of accounts receivable to total assets |
| Inv | The ratio of total inventory to total assets |
| ROA | The ratio of net income to total assets |
| ROE | The ratio of net income to common stockholders' equity |
| Growth | Growth rate in sales in a fiscal year |
| Ins | Percentage of shares held by institutional investors |
| LHR | Percentage of shares held by the largest investor |
| Fee | The log of auditing fee in a fiscal year |
| SOE | Coded one if a firm is state-owned and zero otherwise |
| Big4 | Coded one if a firm is audited by one of Deloitte, Ernst & Young, KPMG, and PricewaterhouseCoopers, and zero otherwise |

*3.3. Models*

First, we developed an OLS regression model (1) to examine the impact of the extent of the digital transformation of enterprises on the establishment of internal control:

$$\text{Establishment}_{i,t} = \alpha + \beta_1 * \text{DT}_{i,t} + \sum \beta * \text{Controls}_{i,t} + \sum \text{Year} + \sum \text{Industry} + \varepsilon \quad (1)$$

Second, we developed an OLS regression model (2) to examine the impact of the extent of the digital transformation of enterprises on the effectiveness of internal control:

$$\text{Effect}_{i,t} = \alpha + \beta_1 * \text{DT}_{i,t} + \sum \beta * \text{Controls}_{i,t} + \sum \text{Year} + \sum \text{Industry} + \varepsilon \quad (2)$$

The dependent variable in model (1) is the establishment of the internal control and the dependent variable in model (2) is the effectiveness of the internal control. The dependent variable is the extent of digital transformation. Additionally, the model controls for time and industry fixed effects using the second-level industry classification from the 2012 industry classification standard of the China Securities Regulatory Commission.

## 4. Results

*4.1. Descriptive Statistics*

Table 4 presents descriptive statistics for the variables. The results indicate that the overall sample mean of internal control establishment is 3.441, with a standard deviation

of 0.394, a minimum value of 0.693, and a maximum value of 4.078. The overall sample mean of the internal control effectiveness is 6.483, with a standard deviation of 0.162, a minimum value of 2.194, and a maximum value of 6.903. These findings suggest that there are significant differences in the establishment and implementation effectiveness of the internal control among various listed companies in China. The descriptive statistics of the control variables are consistent with recent studies, thereby ensuring the reliability and robustness of the research results.

**Table 4.** Descriptive statistics for the main variables.

| Variable | Obs | Mean | Median | Std. Dev. | Min | Max |
|---|---|---|---|---|---|---|
| Establishment | 16,397 | 3.548 | 3.595 | 0.207 | 1.215 | 3.979 |
| Effect | 16,397 | 6.473 | 6.501 | 0.165 | 2.194 | 6.893 |
| DT | 16,397 | 1.394 | 1.099 | 1.603 | 0 | 6.746 |
| Size | 16,397 | 22.143 | 21.992 | 1.169 | 19.885 | 26.003 |
| Age | 16,397 | 2.153 | 2.197 | 0.757 | 0.693 | 3.258 |
| Sub | 16,397 | 2.524 | 2.565 | 0.939 | 0 | 4.997 |
| Lev | 16,397 | 0.416 | 0.409 | 0.194 | 0.055 | 0.878 |
| Sales | 16,397 | 21.471 | 21.333 | 1.348 | 18.450 | 25.481 |
| Cash | 16,397 | 0.713 | 0.143 | 0.115 | 0.016 | 0.608 |
| Rec | 16,397 | 0.124 | 0.104 | 0.099 | 0.002 | 0.466 |
| Inv | 16,397 | 0.124 | 0.106 | 0.096 | 0.008 | 0.532 |
| ROA | 16,397 | 0.037 | 0.035 | 0.052 | −0.261 | 0.188 |
| ROE | 16,397 | 0.056 | 0.064 | 0.110 | −0.863 | 0.302 |
| Growth | 16,397 | 0.281 | 0.135 | 0.587 | −0.673 | 5.053 |
| Ins | 16,397 | 0.429 | 0.449 | 0.239 | 0.003 | 0.903 |
| LHR | 16,397 | 0.339 | 0.321 | 0.140 | 0.087 | 0.733 |
| Fee | 16,397 | 13.811 | 13.710 | 0.709 | 9.210 | 19.403 |
| SOE | 16,397 | 0.351 | 0 | 0.477 | 0 | 1 |
| Opinion | 16,397 | 0.967 | 1 | 0.178 | 0 | 1 |
| Big4 | 16,397 | 0.053 | 0 | 0.223 | 0 | 1 |

### 4.2. Multivariate Analysis

Table 5 presents the relationship between enterprise digital transformation and internal control. The results in column (1) of Table 5 show that the regression coefficient of enterprise digital transformation (DT) on internal control establishment (establishment) is significantly positive at the 1% level. These results support Hypothesis 1 and suggest that when other factors remain constant, the construction of the internal control becomes more effective as the extent of digital transformation in enterprises increases. The results in column (2) of Table 5 show that the regression coefficient of enterprise digital transformation (DT) on internal control effectiveness (effect) is significantly positive at the 1% level. These findings support Hypothesis 2 and indicate that when other factors remain constant, the effectiveness of the internal control is strengthened as the extent of digital transformation in enterprises increases.

**Table 5.** Enterprise digital transformation and internal control.

| Variables | Establishment (1) | Effect (2) |
|---|---|---|
| DT | 0.004 *** | 0.002 *** |
| | (3.65) | (2.62) |
| Controls | Yes | Yes |
| Constant | 2.880 *** | 5.943 *** |
| | (54.69) | (136.32) |
| Year | Yes | Yes |
| Industry | Yes | Yes |

**Table 5.** *Cont.*

| Variables | Establishment (1) | Effect (2) |
|---|---|---|
| Observations | 16,397 | 16,397 |
| Adjusted $R^2$ | 0.244 | 0.165 |

Note: *** indicates significance at the 0.01 level or better, respectively.

### 4.3. Robustness

The impact of the extent of the digital transformation of enterprises on the internal control was tested using a benchmark regression, and the results suggest that digital transformation has a positive impact on internal control. However, since some enterprises did not disclose any keywords related to digital transformation in their annual reports, their inclusion in the sample could have potentially affected the results. To address this issue, we proposed the use of only the enterprises that disclosed digital transformation keywords in their annual reports as research samples, and a regression test was conducted accordingly. The detailed results are shown in Table 6.

**Table 6.** Robustness: remove some samples.

| Variables | Establishment (1) | Effect (2) |
|---|---|---|
| DT | 0.004 *** | 0.003 ** |
| | (2.51) | (1.91) |
| Controls | Yes | Yes |
| Constant | 2.867 *** | 6.022 *** |
| | (41.24) | (100.04) |
| Year | Yes | Yes |
| Industry | Yes | Yes |
| Observations | 8586 | 8586 |
| Adjusted $R^2$ | 0.208 | 0.157 |

Note: ***, ** indicates significance at the 0.01 and 0.05 levels or better, respectively.

The results in column (1) of Table 6 demonstrate that the regression coefficient of enterprise digital transformation (DT) on internal control establishment (establishment) is significantly positive at the 1% level. Similarly, the results in column (2) of Table 6 indicate that the regression coefficient of enterprise digital transformation (DT) on internal control effectiveness (effect) is significantly positive at the 5% level. These findings support the earlier conclusions and suggest that the positive impact of digital transformation on internal control is still valid when only enterprises that disclosed digital transformation keywords are considered.

To ensure the robustness of the research results, we used the proportion of the digital economy-related part of the total intangible assets in the year-end intangible assets disclosed in the financial report notes of listed companies as a measure of the extent of the digital transformation of enterprises [30,31]. Specifically, detailed items were marked as "digital economy technology intangible assets if keywords related to digital technology, such as "software", "network", "client", "management system", and "smart platform" were included in the detailed items. The multiple digital economy technology intangible assets of the same company in the same year were summed up, and the resulting value was divided by the current year's total intangible assets to calculate its proportion.

Table 7 presents the detailed results obtained after re-running the regression test. Column (1) of Table 7 shows that the regression coefficient of enterprise digital transformation (DT) on internal control establishment (establishment) is significantly positive at the 1% level. Column (2) of Table 7 shows that the regression coefficient of enterprise digital transformation (DT) on internal control effectiveness (effect) is significantly positive at the

5% level. These results confirm the previous conclusions and demonstrate the robustness of the findings.

**Table 7.** Robustness: alternative dependent variables.

| Variables | Establishment (1) | Effect (2) |
|---|---|---|
| DT | 0.002 *** | 0.006 ** |
| | (0.65) | (0.28) |
| Controls | Yes | Yes |
| Constant | 2.866 *** | 5.796 *** |
| | (65.65) | (155.93) |
| Year | Yes | Yes |
| Industry | Yes | Yes |
| Observations | 19,475 | 19,475 |
| Adjusted $R^2$ | 0.252 | 0.192 |

Note: ***, ** indicates significance at the 0.01 and 0.05 levels or better, respectively.

To account for the potential time lag between enterprise digital transformation and internal control, and to ensure the robustness of the research results, we treated the dependent variable with a one-period lag. This approach allowed us to consider the time-consuming transmission in enterprise practice and also helped to alleviate potential endogenous problems caused by reverse causality.

Table 8 shows the detailed results of the regression test after lagging the dependent variable of enterprise digital transformation (DT) for one period. Column (1) of Table 8 indicates that the regression coefficient of enterprise digital transformation (DT) on internal control establishment (establishment) is significantly positive at the 1% level. Similarly, column (2) of Table 8 reveals that the regression coefficient of enterprise digital transformation (DT) on internal control effectiveness (effect) is significantly positive at the 10% level. These findings continue to support the previous conclusions, while accounting for the potential time lag in the impact of enterprise digital transformation on internal control.

**Table 8.** Robustness: the independent variable is treated with a one-period lag.

| Variables | Establishment (1) | Effect (2) |
|---|---|---|
| DT | 0.003 *** | 0.001 * |
| | (3.17) | (0.78) |
| Controls | Yes | Yes |
| Constant | 3.618 *** | 4.753 *** |
| | (38.72) | (126.13) |
| Year | Yes | Yes |
| Industry | Yes | Yes |
| Observations | 15,683 | 15,683 |
| Adjusted $R^2$ | 0.278 | 0.153 |

Note: ***, * indicates significance at the 0.01 and 0.10 levels or better, respectively.

### 4.4. Further Analysis

The process of digital transformation varies across enterprises due to the differences in their technical and structural characteristics. To delve deeper into the effects of enterprise digital transformations on internal control, we further divided digital transformation indicators into five sub-items: artificial intelligence technology (AI), blockchain technology (blockchain), cloud computing technology (cloud), big data technology (big data), and digital technology application (application), and tested their impact on the extent of digital transformation on internal control.

Table 9 presents the regression results for the impact of enterprise digital transformation sub-items on the establishment of internal control. The results indicate that the regres-

sion coefficient of artificial intelligence technology (AI) on the establishment of internal control (establishment) is significantly positive at the 5% level. Additionally, the regression coefficient of cloud computing technology (cloud) on internal control establishment (establishment) is significantly positive at the 1% level, and the regression coefficient of digital technology application (application) on internal control establishment (establishment) is significantly positive at the 1% level. However, the regression coefficients of blockchain (blockchain) and big data (big data) technologies on internal control establishment (establishment) are not significant. These findings suggest that the application of artificial intelligence technology, cloud computing technology, and digital technology application in the digital transformation of enterprises significantly improved the establishment of internal control, while the application of blockchain and big data technologies had no impact on internal control establishment.

**Table 9.** Further analysis: impact of enterprise digital transformation sub-items on the establishment of internal control.

| Variables | Establishment | | | | |
| --- | --- | --- | --- | --- | --- |
| | (1) | (2) | (3) | (4) | (5) |
| AI | 0.006 ** | | | | |
| | (2.49) | | | | |
| Blockchain | | −0.003 | | | |
| | | (−0.27) | | | |
| Cloud | | | 0.006 *** | | |
| | | | (2.99) | | |
| Big Data | | | | 0.002 | |
| | | | | (0.45) | |
| Application | | | | | 0.004 *** |
| | | | | | (2.60) |
| Controls | Yes | Yes | Yes | Yes | Yes |
| Constant | 2.876 *** | 2.870 *** | 2.875 *** | 2.871 *** | 2.877 *** |
| | (54.62) | (54.54) | (54.64) | (54.52) | (54.63) |
| Year | Yes | Yes | Yes | Yes | Yes |
| Industry | Yes | Yes | Yes | Yes | Yes |
| Obs | 16,393 | 16,393 | 16,393 | 16,393 | 16,393 |
| Adjusted $R^2$ | 0.243 | 0.243 | 0.243 | 0.243 | 0.243 |

Note: ***, ** indicates significance at the 0.01 and 0.05 levels or better, respectively.

Table 10 presents the regression results of the impact of enterprise digital transformation sub-items on the effectiveness of internal control. The findings show that the regression coefficient of artificial intelligence technology (AI) on the effectiveness of internal control (effect) is significantly positive at the 1% level. Moreover, the regression coefficient of cloud computing technology (cloud) on internal control effectiveness (effect) is significantly positive at the 1% level. However, the regression coefficients of blockchain technology (blockchain), big data technology (big data), and digital technology application (application) on internal control effectiveness (effect) are not significant. These results indicate that the implementation of artificial intelligence and cloud computing technologies in the digital transformation of enterprises has a significant positive impact on the effectiveness of internal control, whereas blockchain technology, big data technology, and digital technology application have no impact on the effectiveness of internal control.

**Table 10.** Further analysis: impact of enterprise digital transformation sub-items on the effectiveness of internal control.

| Variables | Effect | | | | |
|---|---|---|---|---|---|
| | (1) | (2) | (3) | (4) | (5) |
| AI | 0.007 *** | | | | |
| | (3.65) | | | | |
| Blockchain | | 0.000 | | | |
| | | (0.04) | | | |
| Cloud | | | 0.006 *** | | |
| | | | (3.58) | | |
| Big Data | | | | 0.001 | |
| | | | | (0.43) | |
| Application | | | | | 0.002 |
| | | | | | (1.32) |
| Controls | Yes | Yes | Yes | Yes | Yes |
| Constant | 5.944 *** | 5.937 *** | 5.942 *** | 5.938 *** | 5.940 *** |
| | (136.43) | (136.33) | (136.44) | (136.23) | (136.24) |
| Year | Yes | Yes | Yes | Yes | Yes |
| Industry | Yes | Yes | Yes | Yes | Yes |
| Obs | 16,393 | 16,393 | 16,393 | 16,393 | 16,393 |
| Adjusted $R^2$ | 0.165 | 0.165 | 0.165 | 0.165 | 0.165 |

Note: *** indicates significance at the 0.01 level or better, respectively.

We examined the overall impact of enterprise digital transformation on the internal control establishment of listed companies. The findings indicate that enterprise digital transformation has a significant positive influence on internal control establishment.

The DIB internal control and risk management database also constructed indexes by assessing the control environment, risk assessment, control activities, information and communication, and supervisory activities, representing the five elements of internal control, to measure the establishment of the internal control system of the enterprise.

In order to investigate the impact of enterprise digital transformation on the establishment of internal control, we employed model (1) to examine the influence of the extent of digital transformation on the establishment of the control environment (Ctrl-Environ), risk assessment (Risk-Ass), control activities (Ctrl-Act), information and communication (Info-Comm), and monitoring activities (Mon-Act). The findings presented in Table 11 indicate that the regression coefficient of the extent of enterprise digital transformation on risk assessment (Risk-Ass) is significantly positive at the 5% level, while the regression coefficient on control activities (Ctrl-Act) is significantly positive at the 1% level. Furthermore, the regression coefficient of information and communication (Info-Comm) is significantly positive at the 1% level. However, the regression coefficients of the extent of enterprise digital transformation on the control environment (Ctrl-Environ) and monitoring activities (Mon-Act) are not significant. These results suggest that enterprise digital transformation primarily enhances the establishment of internal control in terms of risk assessment, control activities, and information and communication, thereby improving the overall establishment of enterprise internal control.

In addition, we examined the overall impact of enterprise digital transformation on the effectiveness of internal control for listed companies. The results demonstrate that enterprise digital transformation has a significant positive effect on the effectiveness of internal control. To investigate how the digital transformation of enterprises affects the attainment of internal control objectives, we used model (2) to test the impact of the extent of digital transformation of enterprises on the achievement of compliance (CMPL), asset security (SAFE), financial reporting (RPT), operating (OPER), and strategic (STGY) objectives.

**Table 11.** Further analysis: impact of enterprise digital transformation on five elements of internal control.

| Variables | Ctrl-Environ (1) | Risk-Ass (2) | Ctrl-Act (3) | Info-Comm (4) | Mon-Act (5) |
|---|---|---|---|---|---|
| DT | 0.001 | 0.002 ** | 0.011 *** | 0.020 *** | 0.001 |
|  | (0.41) | (1.23) | (3.56) | (9.84) | (0.54) |
| Controls | Yes | Yes | Yes | Yes | Yes |
| Constant | 2.240 *** | 0.347 *** | 1.893 *** | 0.952 *** | 0.626 *** |
|  | (17.73) | (4.52) | (12.74) | (10.05) | (5.75) |
| Year | Yes | Yes | Yes | Yes | Yes |
| Industry | Yes | Yes | Yes | Yes | Yes |
| Obs | 16,393 | 16,393 | 16,393 | 16,393 | 16,393 |
| Adjusted $R^2$ | 0.237 | 0.620 | 0.041 | 0.235 | 0.404 |

Note: ***, ** indicates significance at the 0.01 and 0.05 levels or better, respectively.

The results presented in Table 12 indicate that the extent of the digital transformation of enterprises has a significantly positive impact on asset security objectives (SAFE) at the 10% level and financial reporting objectives (RPT) at the 1% level. The regression coefficient for operating objectives (OPER) is also significantly positive at the 1% level, as well as the regression coefficient for strategic objectives (STGY). However, the regression coefficient for compliance objectives (CMPL) is not significant. These results suggest that digital transformation mainly improves the achievement of asset security, financial reporting, operating, and strategic objectives, thereby enhancing the overall effectiveness of enterprise internal control.

**Table 12.** Further analysis: impact of enterprise digital transformation on objectives of internal control.

| Variables | CMPL (1) | SAFE (2) | RPT (3) | OPER (4) | STGY (5) |
|---|---|---|---|---|---|
| DT | 0.000 | 0.002 * | 0.003 *** | 0.003 *** | 0.013 *** |
|  | (0.15) | (1.82) | (2.75) | (3.21) | (4.51) |
| Controls | Yes | Yes | Yes | Yes | Yes |
| Constant | 6.428 *** | 6.725 *** | 6.180 *** | 5.423 *** | 5.022 *** |
|  | (90.75) | (145.66) | (128.55) | (113.70) | (34.78) |
| Year | Yes | Yes | Yes | Yes | Yes |
| Industry | Yes | Yes | Yes | Yes | Yes |
| Obs | 14,810 | 14,810 | 14,810 | 14,810 | 14,810 |
| Adjusted $R^2$ | 0.077 | 0.155 | 0.213 | 0.311 | 0.134 |

Note: ***, * indicates significance at the 0.01 and 0.10 levels or better, respectively.

*4.5. Heterogeneity Test*

Market competition is a fundamental characteristic of a market economy. In such an economy, companies vie for greater profits and market share driven by their own interests. Through competition, firms expand their resource utilization, improve efficiency, promote the optimal allocation of production factors and resources, ultimately enhancing the overall value of society as a whole.

In a highly competitive product market, managers face increased risks and pressures, and therefore strive to gain a comparative advantage in market competition by improving enterprise management, resource allocation, and adjusting capital structure [35]. Furthermore, product market competition enhances the information environment, leading to the increased stakeholder supervision of corporate management activities, thereby incentivizing firms to strengthen internal governance. However, competition in product markets can also lead to inefficiencies in firms, resulting in lower levels of governance, lower quality financial reporting, and a more opaque information environment. Clinch and Verrecchia [36] found that companies tend to reduce the disclosure of non-mandatory

information due to the potential cost of disclosure, while Li [37] found that corporate information disclosure decreases as product market competition intensifies. Raith [38] argued that fierce product market competition leads to profit fluctuations and increased company-level risks. Consequently, in fiercely competitive product markets, companies tend to adopt looser governance mechanisms to quickly respond to the rapidly changing environment. To test whether the impact of enterprise digital transformation on internal control varies with different market competitions, we divided the samples into two groups based on the median of product market competition, with moderate and intense competitions, respectively.

Table 13 presents the impact of enterprise digital transformation on internal control after grouping based on product market competition. The results in columns (1) and (3) demonstrate that, after controlling for other factors, the influence of enterprise digital transformation on internal control establishment is more pronounced in samples with intense competition. In contrast, the results in columns (2) and (4) suggest that, after controlling for other factors, the impact of enterprise digital transformation on the effectiveness of internal control is limited in samples with relatively moderate competition. However, the positive impact of enterprise digital transformation on internal control is more pronounced in samples with intense competition. In summary, when the product market competition tends to be fierce, the positive impact of enterprise digital transformation on internal control is more pronounced.

**Table 13.** Enterprise digital transformation and internal control: grouping based on product market competition.

| Variables | Moderate | | Intense | |
| --- | --- | --- | --- | --- |
| | Establishment (1) | Effect (2) | Establishment (3) | Effect (4) |
| DT | 0.003 ** | 0.002 | 0.005 ** | 0.003 *** |
| | (2.53) | (1.12) | (2.43) | (2.63) |
| Constant | 2.817 *** | 5.839 *** | 3.000 *** | 6.114 *** |
| | (39.68) | (100.02) | (39.08) | (95.77) |
| Year | Yes | Yes | Yes | Yes |
| Industry | Yes | Yes | Yes | Yes |
| Observations | 8136 | 8136 | 8257 | 8257 |
| Adjusted $R^2$ | 0.255 | 0.171 | 0.232 | 0.163 |

Note: ***, ** indicates significance at the 0.01 and 0.05 levels or better, respectively.

## 5. Conclusions

With the recent advancements in digital technology and the digital economy, the digital transformation of enterprises has become a critical component of traditional industries' transformation. It provides a potential driving force for the innovation and sustainable development of enterprises. Based on the empirical data from A-share listed companies in China's Shanghai and Shenzhen stock markets from 2012 to 2019, we examined the impact of enterprise digital transformation on internal control and drew the following research conclusions.

The digital transformation of enterprises has had a significant impact on the establishment and effectiveness of internal control. In particular, the adoption of artificial intelligence technology, cloud computing technology, and digital technology applications has had a noticeable effect on improving internal control establishment. The use of artificial intelligence and cloud computing technologies has greatly enhanced the effectiveness of internal control.

Furthermore, the digital transformation of enterprises has mainly improved the establishment of internal control in certain areas, such as risk assessment, control activities, and information and communication. This has led to a more robust and efficient internal control system. Additionally, the digital transformation of enterprises has improved the

overall achievement and effectiveness of the enterprise, particularly in areas such as asset security, financial reporting, operating, and strategic objectives.

It is also important to note that the positive impact of enterprise digital transformation on internal control may be limited in relatively moderate market competition. However, when market competition becomes more intense, the benefits of digital transformation on internal control become more pronounced.

## 6. Discussion and Limitation

We presented empirical evidence on the impact of enterprise digital transformation on internal control and offered insights for enterprise managers on how to establish and implement effective internal control systems. First, enterprises should actively respond to the trend of digital technology and economic development. They should leverage the opportunities presented by digital transformation to integrate digital technology and traditional internal control systems, and improve the effectiveness of internal control. Second, the results highlight the positive significance of artificial intelligence technology, cloud computing technology, and digital technology applications in internal control. Enterprises should pay attention to and develop these areas during their digital transformation. Third, the results improve our understanding of the digital transformation of enterprises. We need to explore the role of the degree of digital transformation on risk assessment, control activities, and information and communication. We also need to understand the impact of digital transformation on asset security, financial reporting, operating, and strategic objectives. This will enable us to better integrate digital technology with the establishment and implementation of enterprise internal control. Finally, we provided new ideas for enterprise managers on how to harness the power of enterprise digital transformation in market competition. They need to make good use of digital technology to promote the development of their enterprises.

The previous literature has not investigated the relationship between digital transformation and internal control; we addressed this research gap. However, we did not explore the mechanisms through which the digital transformation of enterprises influences internal control, nor were we able to analyze the economic consequences of this impact on internal control. These limitations represent an area for future research, which would offer a promising direction for further inquiry.

**Author Contributions:** Conceptualization, C.W. and D.W.; formal analysis, C.W. and S.W.; investigation, C.W.; methodology, C.W. and X.D.; resources, C.W.; supervision, D.W.; writing—original draft, C.W. and X.D.; writing—review and editing, D.W. and S.W. All authors have read and agreed to the published version of the manuscript.

**Funding:** This research was funded by Department of Education of Guizhou Province titled "Research on Enterprise Digital Transformation and Internal Control", project number "Scientific Research Project of Department of Education of Guizhou Province (2022) No. 122". This research was funded by Ministry of Education of the People's Republic of China titled "2021 National First-Class Undergraduate Major Construction Points (Financial Management Major at Guizhou Normal University)", project number "General Office of the Ministry of Education (2022) No. 14".

**Institutional Review Board Statement:** Not applicable.

**Informed Consent Statement:** Not applicable.

**Data Availability Statement:** The data will be made available on request from the corresponding author.

**Conflicts of Interest:** The authors declare no conflict of interest.

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
