# Peer review of "Research on the Impact of Enterprise Digital Transformation on Internal Control"

_sustainability, doi:10.3390/su15108392_

Round 1
Reviewer 1 Report
This paper examines the impact of enterprise digital transformation on internal control. However, my comments and suggestions are as follows:
1.More literature should be cited in the introduction section.
2.The contribution is limited.
3.More literature should be added in the section 2. For example, the authors should cite relevant literature in the sentence “Prior researchers have…” (line 117).
4.The time during is from 2007 to 2020. However, the analysis about internal control (Table 1 and Table 2) is from 2012 to 2019. Why not analyze other years?
5. The authors should provide more literature in the sentence “Refer to previous research…” (line 289).
6. The authors measure DT with the total frequency of technical terms. It should be the integer. But in Table 4, the mean of DT is 1.267 with the maximum of 6.806 and the minimum of 0. Did the authors use the natural logarithm?
7.The discussion section should be separate from the conclusion.
8. Are there any limitations of this paper?
9.The authors should revise the language of this paper because there are many minor mistakes in this paper.
Author Response
Dear reviewer,
Thank you very much for your constructive and valuable comments concerning our manuscript entitled " Research on the Impact of Enterprise Digital Transformation on Internal Control". We have learned and benefited a lot from your feedback.
We have read through comments carefully and have made corrections. Based on the comments, we uploaded the file of the revised manuscript. The responses to the reviewer's comments are marked in red and presented following. Please see the attachment. We hope we have addressed your concerns.
Sincerely yours,
All authors

Reviewer 2 Report
The study addresses an important and interesting topic. The theoretical and empirical analyses are well prepared, but the authors have not managed to avoid shortcomings.
(1) Both the executive summary and the introduction lack a clearly formulated objective of the study. This gap needs to be filled.
(2) The theoretical introduction in the paper should be supported by a more reliable and broader review of recent literature in the field of "digital transformation". This gap needs to be filled in order to better justify the importance of the different tools that accompany the digitalisation of the different business areas. The literature should justify more clearly the choice of variables used in the empirical research.
(3) In the conclusion and discussion section, there is a lack of discussion and confrontation of the results obtained with the findings of other researchers on the subject. It is worth filling this gap in order to confirm the validity of the findings.
Overall, the study is very valuable as it identifies factors that are important for the effectiveness of companies in the era of Industry 4.0 and widespread digitalisation. It can inspire strategists on directions and tools to improve the development of modern organisations.
Author Response

(The authors gave the same response as above.)

Reviewer 3 Report
This paper examines the impact of digital transformation on internal control. Overall, the paper is well written and organized with proper literature review and data analysis. Acceptance for publication is recommended.
Author Response
Dear reviewer,
Thank you for acknowledging our work. We have read through comments from other reviewers carefully and have made corrections. We uploaded the file of the revised manuscript. Thank you again for your acknowledging.
Sincerely yours,
All authors
Reviewer 4 Report
The article deals with the impact of corporate digital transformation on internal control. As part of the research, data from companies listed on the stock exchange were analyzed.
Research shows that the extent of digital transformation in companies has a positive impact on the development and effectiveness of internal control. The impact of digital transformation on internal control is limited in companies with relatively moderate market competition, and on the contrary, the positive impact of digital transformation on internal control is more pronounced for companies with stronger competitive markets. Two hypotheses related to the impact of digital transformation on internal control were investigated.
I can state that the methodology used led to relevant results. Research results can lead to a better understanding of digital transformation in order to integrate digital technology and traditional internal control systems and improve the effectiveness of internal control.
Author Response

(The authors gave the same response as above.)
